# 4-1BBL as a Mediator of Cross-Talk between Innate, Adaptive, and Regulatory Immunity against Cancer

**DOI:** 10.3390/ijms22126210

**Published:** 2021-06-09

**Authors:** Alejandra G. Martinez-Perez, Jose J. Perez-Trujillo, Rodolfo Garza-Morales, Maria J. Loera-Arias, Odila Saucedo-Cardenas, Aracely Garcia-Garcia, Humberto Rodriguez-Rocha, Roberto Montes-de-Oca-Luna

**Affiliations:** 1Department of Histology, School of Medicine, Autonomous University of Nuevo Leon, Monterrey 64460, Mexico; alejandra.martinezpr@uanl.edu.mx (A.G.M.-P.); jperez.me0052@uanl.edu.mx (J.J.P.-T.); mdjesus.loeraars@uanl.edu.mx (M.J.L.-A.); odila.saucedocr@uanl.edu.mx (O.S.-C.); aracely.garciagr@uanl.edu.mx (A.G.-G.); humberto.rodriguezrc@uanl.edu.mx (H.R.-R.); 2Internal Medicine Residency Program, University of Texas Rio Grande Valley, Edinburg, TX 78539, USA; rodolfo.garzamorales@utrgv.edu; 3Department of Molecular Genetics, Northeast Biomedical Research Center, Mexican Institute of Social Security (IMSS), Monterrey 64000, Mexico

**Keywords:** 4-1BBL, cancer immunology, immunotherapy, cancer therapy, cancer vaccination

## Abstract

The ability of tumor cells to evade the immune system is one of the main challenges we confront in the fight against cancer. Multiple strategies have been developed to counteract this situation, including the use of immunostimulant molecules that play a key role in the anti-tumor immune response. Such a response needs to be tumor-specific to cause as little damage as possible to healthy cells and also to track and eliminate disseminated tumor cells. Therefore, the combination of immunostimulant molecules and tumor-associated antigens has been implemented as an anti-tumor therapy strategy to eliminate the main obstacles confronted in conventional therapies. The immunostimulant 4-1BBL belongs to the tumor necrosis factor (TNF) family and it has been widely reported as the most effective member for activating lymphocytes. Hence, we will review the molecular, pre-clinical, and clinical applications in conjunction with tumor-associated antigens in antitumor immunotherapy, as well as the main molecular pathways involved in this association.

## 1. Introduction

Cancer incidence cases continue to rise, registering over 18 million recent cases and over 9 million deaths in 2018, distributed worldwide [1]. Current therapeutic options include surgery, radiation therapy, and chemotherapy, which, despite being efficient for tumor clearance, have significant side effects that affect the patient’s quality of life. They have also registered a high recurrence rate and the formation of tumor-resistant clones [2,3]. Hence, new therapeutic approaches are essential to counteract such inconveniences and to ensure full tumor cell eradication.

Gene therapy involves using genetically modified RNA and/or DNA coding homologous or heterologous proteins to replenish a specific protein deficiency or trigger a specific cell response. This is the most used strategy to express antigens and immunostimulatory molecules to induce protective memory immunity against various pathogens and tumor cells [4].

4-1BBL is an immunostimulant molecule that interacts with the 4-1BB high-affinity receptor during the antigen presentation, providing costimulatory signals to both CD4+ and CD8+ T cells through the activation of NF-kB, c-Jun, and p38 downstream pathways, triggering pleiotropic effects on the immune system [5]. Thus, it has been proposed to use it as a complex with epitopes or antigens of pathogens that induces an increase in the specific immune response of antigen [6].

Cancer development can result from spontaneous cell mutations or viral oncoproteins that trigger cell malignancy processes; therefore, on tumor cells, we can detect proteins that can serve as antigens for cancer therapy. Cancer cells express several tumor antigens. According to their exclusive expression on tumor cells or tumor and non-tumor cells, they are classified as tumor-specific antigens (TSAs) or tumor-associated antigens (TAAs), respectively [7].

TAAs are over-expressed in cancer cells, and because of this, they have been used in the diagnosis, prognosis, and treatment of cancer. Alpha-fetoprotein (AFP), carcinoembryonic antigen (CEA), cancer antigen (CA), tissue polypeptide-specific antigen (TPS), and prostate-specific antigen (PSA) are some currently used examples [8].

Particularly, Bruggen et al. identified, three decades earlier, a gene encoding an antigen expressed in several human melanoma tumors. Cytolytic T lymphocytes recognize this antigen, which marked a guideline in specific immunotherapy against cancer [9,10]. The immunogenicity triggered by such antigens is limited because cancer cells can develop several mechanisms to evade the immune system, such as selecting tumor cell variants that lose antigen expression and/or decrease the expression of MHC molecules [11]. Focused on these findings, there are developing strategies with the use of tumor antigens attached to adjuvant molecules for the improved delivery of in vivo antigens. In this way, they generate a more efficient antitumor immune response [12,13,14].

## 2. 4-1BB/4-1BBL

### 2.1. Receptor 4-1BB

4-1BB was first identified in mice as an inducible mRNA sequence found in cytotoxic T-lymphocytes and a helper T-lymphocyte clone by a modified differential screening procedure during the isolation of putative T cell-specific expression genes [15]. Sequence analysis of the 4-1BB gene exhibited similarity to members of the TNF family [16]. This murine 4-1BB (m4-1BB) maps in chromosome 4 close to the p80 form of the tumor necrosis factor receptor and the gene for CD30 [17]. The human homolog of 4-1BB (h4-1BB) (CD137) was cloned from activated human T-cell leukemia virus type 1-transformed human T-lymphocytes library [17], and comprises 255 a.a. maps to chromosome 1p36, sharing a 60% identity with murine 4-1BB [18].

Both m4-1BB and h4-1BB consist of a type I transmembrane receptor with four extracellular cysteine-rich domains, followed by a short transmembrane domain and a C-terminal cytoplasmic domain that is necessary for the binding of adaptor proteins to facilitate signaling [19,20].

4-1BB expression as an inducible receptor depends on the activation of T cells by some agonists such as plate-bound anti-CD3, concanavalin A, phytohemagglutinin, interleukin (IL)-2, IL-4, CD28, phorbol myristyl acetate, or ionomycin in the presence of antigen-presenting cells (APC). It is expressed in CD4+, CD8+, natural killers (NK), NKT, and constitutively in CD11c+ dendritic cells (DCs) and CD4+ CD25+ regulatory T cells [21].

### 2.2. 4-1BB Ligand

The presence of a ligand for 4-1BB was first verified in the EL4 murine cell line through its attachment with a fusion 4-1BB/Fc protein. The 4-1BBL gene was then cloned through the screening of an EC1 cDNA expression library [22]. 4-1BBL is expressed in several APCs, such as B lymphocytes, macrophages, and DCs, and activated T cells [23].

Murine 4-1BBL (m4-1BBL) is a type II transmembrane protein composed of an N-terminal cytoplasmic region and a C-terminal ectodomain separated by a transmembrane domain. We can divide the ectodomain into a tail region and the TNF homology domain (THD). The latter handles the interaction with its cognate receptor m4-1BB. m4-1BBL is self-assembled as a two-fold symmetrical homodimer, in which a disulfide bond covalently connects both protomers [24].

Human 4-1BBL (h4-1BBL) was first isolated using a fusion protein consisting of the extracellular portion of h4-1BB coupled to the Fc region of human immunoglobulin (Ig) G1 to identify and clone the gene for h4-1BBL from an activated CD4+ T-cell clone using a direct expression cloning strategy. Sequence analysis revealed that h4-1BBL comprised 254 a.a. and shared a 36% identity with murine 4-1BBL [18].

h4-1BBL exists both in soluble form and as a cell-bound type II transmembrane protein that comprises a short N-terminal cytoplasmic region, followed by a transmembrane domain, and extracellular TNF THD, which binds to 4-1BB. The THD region of h4-1BBL shares just 20–25% of sequence identity with other recognized TNF ligands. However, the overall structure of h4-1BBL is like other non-covalent homotrimeric TNF ligands [20].

### 2.3. 4-1BB/41BBL Complex

All the characterized conventional human or murine TNF ligands organize into a symmetrical trimeric bell-shape to form a functional biological unit [19]. The structure of h4-1BBL also assembles as a trimeric bell-shape despite its low-sequence similarity with conventional members [25]; however, m4-1BBL, although it contains most of the characteristic features required for packing into a bell-shape, is assembled in an atypical dimeric structure [24].

m4-1BBL and h4-1BBL protomers are structurally similar and share structural details with members of the conventional TNF family. The major difference occurs in the oligomeric assembly of the ligand because human 4-1BBL forms a hexameric functional unit, and m4-1BBL forms a tetrameric signaling unit [24]. Such structures supply insights into the structural differences that drive the species-specific receptor–ligand interactions. Since multimerization and clustering is a precondition requirement for TNFR intracellular signaling, the presentation of differential functional signaling units in human and mouse supports a unique mechanism of 4-1BB signaling in both species [26].

Comparable to other TNF members, the association with 4-1BBL oligomerizes 4-1BB, recruiting intracellular trimeric TRAFs (TRAF1 and TRAF2), which lead to stimulation of NF-κB and phosphatidylinositol 3-kinase/Akt-mediated proinflammatory signaling pathways [27] (Figure 1).

### 2.4. SA-4-1BBL

4-1BBL as a soluble trimeric molecule displayed the absence of biological activity [28]; thus, a soluble chimeric SA-4-1BBL was generated that consists in the fusion of the extracellular domain of 4-1BBL to the C-terminus of a changed form of core streptavidin (SA). Such a molecule exists as tetramers and oligomers owing to the structural features of SA and can cross-link 4-1BB receptors, which are capable of inducing potent activation of CD4+ and CD8+ T cell activation [29] (Figure 1). The soluble form of the 4-1BB ligand has proved to be a better immunostimulant with fewer adverse effects than 4-1BB receptor agonistic monoclonal antibodies [30].

An unexpected feature of SA-4-1BBL has recently been demonstrated; moreover, the known effect in CD8+ T-effector and memory response, non-specific activation of CD4+ T and NK cells, exhibited protection against tumor challenge [31], depicting a bridge between innate and adaptive immune response.

## 3. Tumor-Associated Antigens (TAAs)

The antigen MZ2-E, identified in melanoma tumor cells by Bruggen et al., was also expressed in other melanoma cell lines and different histological types, but without expression in normal tissues. The gene coding this antigen was detected silent or quasi-silent in most normal tissues and activated in the tumors, even when the gene sequence appeared identical in normal tissues and tumors. In addition, these antigens were recognized by killer T lymphocytes [9]. In recent decades, similar antigens have been widely used as components of antitumor vaccines.

Vaccines that rely on antigen specificity offer the greatest advantage compared to other non-specific conventional therapies such as tumor resection, radiotherapy, and antitumor chemotherapy [32]. However, given the existence of resistance mechanisms in tumor development such as a loss or change of epitopes recognized by immune cells, T cell exhaustion, antigen tolerance, and the infiltration of immunosuppressive cells [11], the need arises to associate these antigens with molecules that improve the immune response [33].

### SA-4-1BBL and TAAs

Four distinct nodes can induce antitumoral immunity: eradicating the immune suppression in the tumor microenvironment, inducing immunogenic cancer cell death, enhancing the APC function/adjuvanticity, and enhancing T/macrophage effector activity [34]. The combination of the SA-4-1BBL adjuvant with TAA results in the activation of the previous four mechanisms.

Typically, a subset of ligands of tumor necrosis factor receptor (TNFR) superfamily members induces the non-canonical NF-κB pathway in immunity and inflammation [35]. In addition to the recognized functions of this pathway, including the regulation of lymphoid organ development, B cell survival, and maturation [36,37], an important role in regulating DC development and maturation has been reported using mice deficient in NF-B members RELB, cREL, or both cREL and NF-B1, noting that DCs require RELB to induce T cell responses via both the conventional antigen-presenting pathway and via cross-priming [38,39,40], which implies an essential role in the anti-tumor immune response [41].

In contrast to the rapid and transient activation of the canonical NF-κB pathway, the activation of the non-canonical NF-κB pathway is characteristically slow and persistent [40], implying major specificity.

In terms of eliminating immune suppression, there has been a significant reduction in the frequency of Treg cells following vaccination with SA-4-1BBL/MPL as the adjuvant component of E7 TAA-based vaccine against TC-1 tumors in a C57BL/6 cancer mice model compared to PBS or E7 alone controls [42]. Regarding the induction of immunogenic cancer cell death, it has been reported as vaccines with E7 plus SA-4-1BBL that generate a CTL and Th1 response by augmented memory pool for both CD4^+^ and CD8^+^ T cells and improved T cell proliferative, killing, and Th1 cytokine responses in long-term surviving mice [13,42,43,44]. Adjuvants that promote APC function are well-known to enhance Th1 cell or M1 macrophage effector pathways, activating both nodes 3 and 4 [34]. Immunization with a DNA vaccine encoding SP-SA-E7-4-1BBL in vivo showed an increase in antigen-specific interferon (IFN) levels associated with the therapeutic efficacy as Th1-mediated response to tumor eradication [43]. To improve the administration of SP-SA-E7-4-1BBL, we constructed an oncolytic adenovirus that encodes the previous construction and showed a specific antitumor effect in an established tumor mouse model [45].

## 4. Tumor Microenvironment

Tumor microenvironment regulation has a key role in the success of antitumor therapies. Composed of proliferating tumor cells, tumor stroma, blood vessels, infiltrating inflammatory cells, and a variety of associated tissue and molecules cells, it emerges dynamically in the process of tumor growth because of its interactions with the host [46].

The cross-talking orchestrated by tumor cells and their microenvironment includes contacts such as cell to cell, cell-free structures, and soluble mediators. Therefore, tumor cells can control the microenvironment, making the non-malignant cells present to work for their advantage, leading to immune evasion and tumor progression [47,48].

To achieve effective antitumor immunity, first, DCs must take antigens released from the tumor simultaneously with antigen uptake, and DCs must also receive an activation signal from factors released from dying tumor cells. Next, DCs loaded with tumor antigens must migrate to the lymphoid organs to generate antigen-specific CD8+ cytotoxic effector cells. Finally, specific antigen cancer T cells must infiltrate the immunosuppressive microenvironment [49].

The immunosuppression state is produced by downregulating major histocompatibility complex class I (MHC I) or expressing cell surface molecules such as programmed cell death protein 1 ligand 1 (PDL1) by tumor cells and the release of immunosuppressive molecules such as transforming growth factor-β (TGFβ), indoleamine 2,3-dioxygenase (IDO), arginase, and nitric oxide synthase as a consequence of a hypoxic environment [49,50].

## 5. Delivery Technologies for 4-1BBL

The delivery system is a crucial point for the success of antitumor therapy, since access to the tumor tissue will depend on it. Therefore, a thorough analysis of the treatment used, the type of tumor, and the administration route are necessary to ensure the best possible delivery. Herein, we review five dominant strategies used in the 4-1BBL adjuvant delivery in the preclinical and clinical trials reviewed.

### 5.1. Fusion Proteins

The design of chimeric fusion proteins that express different genes in a single peptide offers the advantage of simultaneously targeting several signaling pathways, and this is a promising alternative to antibody therapy, with minor secondary adverse effects. In recent years, fusion proteins have increased in cancer immunotherapy [51,52,53]. DSP107 is a new immunotherapeutic fusion protein, termed Dual Signaling Protein 107. It combines innate and adaptive immune response activation by blocking CD47/SIRPα interaction and activating 4-1BB. This fusion protein contains the extracellular domains of SIRPα and 41BBL. 41BBL allows protein trimerization, which is essential for 4-1BB receptor activation. DSP107 binds to CD47 with subsequent removal of the inhibitory signal delivered to phagocytes on tumor cells. CD47 binding of DSP107 allows the delivery of the 4-1BBL costimulatory signal to tumor local T-cells [54].

Another example of employing such technology is a preclinical study that used a bifunctional fusion protein derived from tumstatin and 4-1BBL (rh4TFP), with the aim of targeting both angiogenesis and T cell activation. The protein comprises a T7 peptide from inhibitor tumstatin and the extracellular domain of 4-1BBL, both coupled with different linkers. rh4TFP-2 exhibits antiangiogenic activity similar to tumstatin by inhibition of proliferation and migration of human umbilical vein endothelial cells. Additionally, it exhibits an increase of T lymphocyte activation for the release of IL-2 and IFN-γ, resulting in T lymphocyte activation by 4-1BBL. It shows tumor growth suppression and prolonged survival in a B16F10 melanoma-bearing mouse model [53].

4-1BBL fused to tumor-associated antigens as peptide-based vaccines also show a potent activation of the innate, adaptive, and regulatory immune response in established tumor mice models [44,55].

### 5.2. DNA Vaccines

Therapy based on DNA administration offers the advantage of being a simpler process if we compare it with the elaboration of fusion proteins. To simplify the system of fusion proteins, a chimeric HPV-16 E7 DNA vaccine (SP-SA-E7-4-1BBL) was generated that contains the signal peptide (SP) of calreticulin (CRT), the streptavidin (SA) domain of SA-4-1BBL, HPV-16 E7 double mutant gene, and the extracellular domain of mouse 4-1BBL. It shows prophylaxis against the TC-1 tumor, a therapeutic effect against an established TC-1 tumor, and an increased frequency of E7-specific T cells producing IFN-γ [43]. However, because of the local administration of DNA vaccines, transfection is limited to cells adjacent to the injection site, with the consequent need for multiple immunizations to increase an effective antitumor effect [56,57].

### 5.3. Oncolytic Virus

Oncolytic virotherapy is encouraging for antitumor gene therapy as it can selectively replicate in tumor cells, causing cell lysis. Additionally, tumor cell debris is released and taken in by immune cells to improve antitumor responses. An oncolytic adenovirus (OAd) expressing SP-SA-E7-4-1BBL was competent in infecting murine cancer and normal cells, and the expressed protein was able to target the endoplasmic reticulum (ER). Moreover, this OAd caused a cell-killing effect specific to cancer cells and generated a specific antitumor effect in vivo. Administration of OAd in mice with established TC-1 tumors resulted in tumor growth suppression and 100% survival when contrasted with the reference positive control. However, additional studies should analyze the safety and biodistribution of recombinant adenovirus and associate the mechanisms implicated in the antitumor effect [45].

### 5.4. Antibodies

Specific antibodies against 4-1BB alone or in combination with other agents are being studied and developed to activate and enhance anti-cancer immune responses. There are ongoing clinical trials to evaluate the efficacy of agonistic 4-1BB antibodies alone or in combination with other treatment modalities. However, treatment with 4-1BB agonist antibodies has resulted in adverse events such as fatal liver toxicity [58,59]. 4-1BB agonist antibodies such as urelumab (BMS-66513) and utomilumab (PF-05082566) are now being studied at lower doses as monotherapy and in combination with other anti-cancer agents [60]. Mixture therapies of anti-CD137 with other antibodies or other reagents have exhibited great potentials in anti-tumor activities and reduced the probability of systemic toxicities. More clinical advancement is required to fully unlock the use of this antibody [61].

### 5.5. Cellular Therapy

Early adoptive immunotherapy primarily transfuses autologous or allogeneic tumor-responsive T cells back into the patient’s body to damage the patient’s tumors. CAR-T cell therapy and T cell receptor (TCR)-T cell therapy are the most used and most effective immunotherapy technologies. CAR-T cells are T-cells genetically changed to express a synthetic construct comprising of a synthetic T-cell receptor (TCR) targeted to a programmed antigen expressed on a tumor [62]. Moreover, by “armoring” these cells with genetic modifications such as IL-12, CD40L, or 4-1BBL, it produces signaling cascades similar to their normal counterparts and enhances T cell activation, expansion efficacy, and persistence [63]. Pre-clinical studies with T-cells co-transduced with 4-1BBL and CD80 showed robust proliferation and increased cytokine production compared with T-cells transduced with either construct alone [64]. CAR-T cells co-transduced with 4-1BBL have also shown enhanced in vitro and in vivo efficacy in a mouse systemic tumor model [65]. An alternative cellular approach involves using induced pluripotent stem (iPS) cell-derived myeloid lines (iPS-ML). The benefits of using iPS-ML are their infinite proliferative capability and ease of genetic modification. Kuriyama et al. demonstrated that peritoneal injections of iPS-ML-41BBL significantly impede the tumor growth of peritoneally disseminated melanoma and extended survival compared to that of iPS-ML in a mouse melanoma model [66].

## 6. 4-1BBL Current Clinical and Preclinical Applications

Cancer therapies that focus on stimulating a specific immune response in combination with adjuvants that modulate such response could counteract said immunosuppression by seeking the predominance of a pro-inflammatory state in the tumor microenvironment [34,67,68,69]. In the search for potent anti-tumor immunotherapy with minor toxic side effects, advanced materials and new and more efficient delivery systems are being developed [70,71,72].

We should note that tumors represent a hard target because modulation of the tumor immune microenvironment is necessary to achieve successful therapy. Preclinical and clinical studies explore several strategies that involve different molecular pathways to reach this.

Using SA-4-1BBL in preclinical models has shown a protector effect and a therapeutic effect in mice with established tumors. This effect increases the specific long-term immune response required by IFN-γ as the mediator of cross-talking between innate and adaptive immunity [31,43,44] (Table 1). This protective effect also influences the regulatory immunity system by promoting a favorable intra-tumoral environment determined by the CD8+ T eff/CD4+ Foxp3+ Treg cell ratio [42].

To date there are no records regarding the use of the SA-41BBL adjuvant in clinical trials; however, there are several currently using 4-1BBL as monotherapy or associated with different molecules (Table 2).

### 6.1. DSP107

The most recent, DSP107 (SIRPα–4-1BBL), is a bi-functional, trimeric fusion protein. This study evaluates the safety, pharmacokinetics (PK), and pharmacodynamics (PD), as well as the first evaluation of the efficacy of DSP107 as monotherapy or in combination with atezolizumab. The study was designed in two parts: the first as monotherapy dose-escalation of DSP107, which comprises intravenous administration in patients with advanced solid tumors, as long as they are not amenable to any beneficial therapeutic options. In addition, the study includes a cohort to establish a safe dose when DSP107 is administered in combination with atezolizumab in the second part of the study. This last part of the study will include participants with non-small-cell lung cancer who have shown progression after first-line treatment with PD-1- or PD-L1-targeting agents, and have formerly achieved a better response or stable disease.

Another of the studies posted in clinical trials last year was RTX-240, a cellular therapy of engineered red cells that co-expressed 4-1BBL and IL-15TP, a fusion of IL-15 and IL-15 receptor alpha that harnesses the innate and adaptive immune systems to treat cancer. This study is a phase 1/2 multicenter, multidose, first-in-human (FIH) dose-escalation trial and expansion to determine the safety and tolerability.

### 6.2. LOAd703

An oncolytic adenovirus armed with a transgene encoding TMZ-CD40L and 4-1BBL can selectively lyse tumor cells, showing a capacity to induce anti-tumor cytotoxic T-cell responses, and reduce myeloid-derived suppressor cell (MDSC) infiltration and tumor regression in a preclinical study [73]. Subsequently, in a phase I/II trial, patients with unresectable or metastatic pancreatic ductal adenocarcinoma (PDAC) were treated with LOAd703 intra-tumoral injections and standard nab-paclitaxel/gemcitabine (nab-P/G) chemotherapy.

Three subjects received dose 1 (5 × 10^10^ VP), 4 subjects received dose 2 (1 × 10^11^ VP), and 6 subjects received dose 3 (5 × 10^11^ VP). The most frequent adverse events (AEs) attributed to LOAd703 were fever, chills, nausea, and increased transaminases. Such AEs have been transient and grade 1–2, except for a grade 3 transaminase elevation in 1 subject receiving dose 3. Throughout the protocol therapy, circulating MDSCs were reduced in 8/13 subjects, whereas effector memory T-cells were augmented in 10/13. ELISPOT analyses exhibited an increase in tumor antigen-specific T-cells in 10/13 subjects. Next to the lowest dose level, the best response was stable disease, and 6/10 patients who received higher LOAd703 doses had a partial response. Only 1 patient has had progressive disease as the best response [74].

LOAd703 is also evaluated in a phase I/II trial investigating intra-tumoral treatments of virus combined with intravenous infusions of atezolizumab in malignant melanoma.

### 6.3. RO7227166

RO7227166 evaluates the safety, tolerability, and efficacy of a CD19-targeted 4-1BB ligand in a phase I, open-label, dose-escalation study in subjects with relapsed/refractory non-Hodgkin’s lymphoma (part I and II), and follicular lymphoma and diffuse large B-cell lymphoma (part III). This ligand was administered intravenously in combination with obinutuzumab and glofitamab. This study was divided into three parts, a dose-escalation stage (part I and II) and a dose-expansion stage (part III). Part I comprised administering a fixed-dose obinutuzumab seven days prior to the first administration of RO7227166, followed by an intravenous infusion in a combination of both in a three-weekly schedule. In part II, subjects received a fixed dose of obinutuzumab seven days before the administration of RO7227166, followed by a combination of the latter with glofitamab in a three-weekly schedule. The part III dose-expansion stage participants will receive RO7227166 combined with glofitamab by intravenous infusion in a three-weekly schedule.

### 6.4. EGFRt/19-28z/4-1BBL CAR-T Cells

EGFRt/19-28z/4-1BBL CAR-T cells comprise T cell enrichment, activation, and genetic modification by a retroviral vector encoding a CD19-targeted CAR, the co-stimulatory ligand 4-1BBL, and the EGFRt safety system (EGFRt/19-28z/4-1BBL) from peripheral blood of subjects that have relapsed or refractory chronic lymphocytic leukemia (CLL). This phase I study aims to assess the safety of several dose levels of these prepared cells obtained from the subjects and to obtain a safe dose of these T cells for patients with this type of condition that has progressed after traditional therapy. Additionally, they want to discover what effects these modified T cells show on the patient and cancer. Modified T cell infusions will be administered for 2–7 days, with the subsequent conclusion of the treating investigator’s choice of preparing chemotherapy. They will perform subsequent treatment with serial blood and bone marrow sampling to evaluate toxicity, therapeutic effects, and survival of the genetically changed T cells. Each cohort of 3–6 patients will be treated with increasing doses of modified T cells. At a minimum, 3 subjects will be treated at each dose level with an accumulation of only 2 patients monthly within each dose level. At any rate, two weeks will pass by from the first T cell infusions before the second patient is treated (on dose level 1) to set a limit for toxicity and safety evaluation. Every patient treated at the previous dose level will be monitored a minimum of 4 weeks before dose escalation. There are 4 scheduled dose levels: 1 × 10^5^, 3 × 10^5^, 1 × 10^6^, and 3 × 10^6^ CAR-T cells/kg.

### 6.5. HLA A2/4-1BB Ligand

A melanoma vaccine changed to express the HLA A2/4-1BB ligand consists of a cell line with a high expression level of melanoma molecules and is genetically modified to generate a robust immune response. The study is based on the hypothesis that stimulation of the immune response against the tumor can better destroy residual tumor in melanoma patients with a very high risk for disease recurrence and in patients with a comparatively low tumor burden who previously received first-line therapy for their disease. Previous clinical trials have shown that vaccination of subjects with a cell line of tumor cells from the patient themself, or with a mixture of three cell lines that partly match the patient’s cell characteristics, could enhance the immune response against the tumor and was linked with improved disease-free and overall survival.

## 7. Conclusions

There is still a long way to go in the battle against cancer. However, more knowledge is available regarding how to defeat it. Stimulating the immune system has historically proven to be a double-edged sword; therefore, it is imperative to find a balance between attacking the enemy and causing less damage to the host. According to the results analyzed in this review, the use of 4-1BBL in preclinical trials has shown an efficient activation of the innate, adaptive, and regulatory immune system, turning the tumor microenvironment into a propitious place to generate an efficient, specific, and lasting immune response. Currently, the efficacy of this response has been shown in clinical trials. Preliminary results show a safe molecule with minor adverse effects that induces an efficient activation of the immune system against the tumor. However, further research is needed to establish this antitumor therapy.

## Figures and Tables

**Figure 1 ijms-22-06210-f001:**
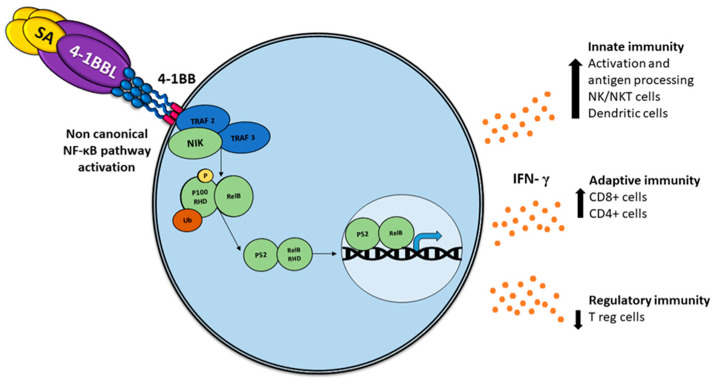
Activation of the non-canonical NF-κB pathway by SA-4-1BBL through the tumor necrosis factor receptor (TNFR) includes the slow and persistent activation of the NF-κB-inducing kinase (NIK), phosphorylation of NIK-mediated p100, and subsequent nuclear processing and translocation of p100 from p52 and RELB, with the consequent production of interferon (IFN), which triggers pleiotropic effects on the immune response.

**Table 1 ijms-22-06210-t001:** Preclinical studies with the adjuvant SA-4-1BBL.

Approach	Findings	Immune Signaling Pathway	Year, Ref.
Prophylactic and therapeutic effect of DNA vaccine in a cervical cancer mouse model.	Prophylaxis against TC-1 tumor. Therapeutic effect against established TC-1 tumors.	Increased frequency of E7-specific T cells producing interferon IFN-γ.	2019 [43]
Tumor protection of subunit vaccine in three different tumor types (TC-1, LLC, or 3LL-huMUC1) in a mouse model.	Monotherapy protects mice against tumor challenge. A rapid and lengthy window of protection against the tumor. Protection is tumor-type-independent and does not evolve into a long-lasting immune memory. Prevents post-surgical tumor recurrence.	IFN-γ+ producing CD4+ T and NK cells as predictors of SA-4-1BBL-mediated immune protection against tumors. Protection against the tumor requires IFN-γ as a mediator of crosstalk between NK and CD4+ T cells.	2019 [31]
Vaccine adjuvant system effect in a mouse model of human papilloma virus (HPV)-induced cancer.	SA-4-1BBL/MPL as the adjuvant component of the E7 TAA-based vaccine has robust efficacy in eradicating established TC-1 tumors. SA-4-1BBL/MPL controls 3LL pulmonary metastasis progression. Therapeutic efficacy of the SA-4-1BBL/MPL is achieved in the absence of autoimmunity and detectable clinical toxicity.	The therapeutic efficacy of SA-4-1BBL/MPL is associated with a robust effect of SA-4-1BBL and MPL on the generation of peripheral CD8+ T cell responses. Vaccination with the SA-4-1BBL/MPL results in a favorable intra-tumoral CD8+ T eff/CD4+ Foxp3+ Treg cell ratio. CD8^+^ T cells and IFN-γ are critical to the therapeutic efficacy of SA-4-1BBL/MPL while Treg cells are detrimental to the efficacy of MPL monotherapy.	2014 [42]
Therapeutic efficacy of subunit vaccine in 3LL lung carcinoma in mice.	Eradicating 3LL tumors in mice.	The therapeutic efficacy of the vaccine is associated with robust CD8+ T cells and NK cells’ effector responses. CD8^+^ T cells play an obligatory role, while NK cells play a moderate, but significant, role in the therapeutic efficacy of the vaccine.	2012 [44]
Therapeutic efficacy of a protein-based vaccine in a mouse cervical cancer model.	Eradication of established TC-1 tumors and generates long-term tumor-specific memory response.	Vaccination with E7 protein and SA-4-1BBL generates primary T cell responses. Generation of robust T cell proliferative and effector responses. Long-term T cell memory pool and enhanced intra-tumoral CD4+ and CD8+ T cells. NK cells play a critical role in vaccine efficacy.	2010 [13]

**Table 2 ijms-22-06210-t002:** Current Clinical trials with -4-1BBL.

Intervention Model	Year; Phase
An open-label, multicenter, multidose, first-in-human study of RTX-240 for the treatment of patients with relapsed/refractory R/R or locally advanced solid tumors. RTX-240 is a cellular therapy that co-expressed 4-1BBL and IL-15TP, a fusion of IL-15 and IL-15 receptor alpha as monotherapy.	2020; I, II
Study of DSP107 in subjects with advanced solid tumors including a dose-escalation safety study (part 1) and preliminary efficacy assessment of DSP107 as monotherapy and in combination with atezolizumab (part 2). DSP107 (SIRPα–4-1BBL) is a bi-functional, trimeric, fusion protein.	2020; I, II
LOAd703 in combination with atezolizumab in malignant melanoma. LOAd703 is an oncolytic adenovirus encoding trimerized membrane-bound (TMZ)-CD40L and 4-1BBL.	2019; I, II
An open-label study to evaluate the safety, pharmacokinetics, and preliminary antitumor activity of RO7227166 (a CD19-targeted 4-1BB ligand) in combination with obinutuzumab and in combination with glofitamab following a pre-treatment dose of obinutuzumab administered in participants with relapsed/refractory B-cell non-Hodgkin’s lymphoma.	2019; I
Evaluating the effect of LOAd703 in patients with pancreatic cancer, biliary cancer, ovarian cancer, and colorectal cancer. LOAd703 is an oncolytic adenovirus serotype 5/35 encoding immunostimulatory transgenes: TMZ-CD40L and 41BBL.	2017; I, II
A CD19-targeted EGFRt/19-28z/4-1BBL “armored” Chimeric Antigen Receptor (CAR) modified T cells in patients with relapsed or refractory CD19+ hematologic malignancies.	2017; I
Evaluating safety of LOAd703, an armed oncolytic adenovirus for pancreatic cancer. Delolimogene mupadenorepvec oncolytic virus encoding TMZ-CD40L and 4-1BBL.	2016; I, II
Allogeneic vaccine modified to express HLA A2/4-1BB ligand for high-risk or low residual disease melanoma patients.	2013; I, II

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
