# Peer review of "4-1BBL as a Mediator of Cross-Talk between Innate, Adaptive, and Regulatory Immunity against Cancer"

_ijms, 2021, doi:10.3390/ijms22126210_

Round 1

Reviewer 1 Report

Nice paper, well written and significantly contributing to scientific knwoledge

Author Response

We thank you very much for your comments and the time employed in the manuscript revision.

Reviewer 2 Report

The language is poor and it requires an extensive revision before being considered for publication. Moreover it is often challenging to judge the scientific contribution of the present review due to several difficult to understand sections.

Author Response

We sincerely appreciate your comments. To address this situation, we sent the manuscript to an English editing service.

Reviewer 3 Report

In this manuscript Alejandra G. Martinez and coauthors review the function, structure and therapeutic use of 4-1BBL. The authors describe the structure of the 4-1BB receptor, the interaction with its ligand 4-1BBL and its molecular downstream effects. The authors describe the current uses of 4-1BBL as costimulatory molecule in cancer immunotherapy and tumor vaccination. The authors also describe distinct preclinical and clinical applications that involve 4-1BBL or specific antibodies as monotherapy or in combination. Moreover, the authors indicate distinct delivery methods used to employ 4-1BBL therapeutically while giving a wide understanding of the pro and cons of each approach.

The manuscript is timely and interesting. It reads well, however there are many typos and grammatical mistakes than should be corrected. The manuscript quotes many interesting studies that employ 4-1BBL. Altogether it is a good review manuscript.

Comments

Although the review reads well, some parts of the text are unclear, and the authors should correct them. Some of these parts are:

  • Line [57] to [63]
  • Line [150] to [153]
  • Line [197] to [203]
  • Line [215] to [220]

As the preclinical studies are not well described in the text as the clinical ones, I suggest adding to Table 1 a “strategy” column, where briefly it is described the context and the strategy used in these experiments. This could help the reader to better understand the findings. For example, it will increase the relevance of the paper adding tumor type, mouse model and delivery system.

Section 6 (Delivery technologies for 4-1BBL) describes the structure of most of the drugs used in the reviewed clinical trials, thus I suggest moving section 6 before section 5. This could help the reader to better understand the 4-1BBL-based therapeutic interventions that have been investigated.

The description for table 2 is missing.

Human and mouse descriptions of 4-1BB are identical, authors could fuse these in one paragraphs.

Capital letters should be used only when necessary.

Author Response

We appreciate the time and effort that you have dedicated to providing your valuable feedback on our manuscript.

Comments

Point 1. Although the review reads well, some parts of the text are unclear, and the authors should correct them. Some of these parts are:
We rephrase these sections as follows:

Line [57] to [63] (lines 57-63)

Line [150] to [153] (lines 147-152)

Line [197] to [203] (lines 199-201)

Line [215] to [220] (lines 306-311)

Point 2. As the preclinical studies are not well described in the text as the clinical ones, I suggest adding to Table 1 a “strategy” column, where briefly it is described the context and the strategy used in these experiments. This could help the reader to better understand the findings. For example, it will increase the relevance of the paper adding tumor type, mouse model and delivery system.
We added a column to table 2 entitled “approach” in which is displayed delivery system, tumor type and in vivo model.

Section 6 (Delivery technologies for 4-1BBL) describes the structure of most of the drugs used in the reviewed clinical trials, thus I suggest moving section 6 before section 5. This could help the reader to better understand the 4-1BBL-based therapeutic interventions that have been investigated.
Agreeing with your opinion to improve the readability of the manuscript, we moved the section (Delivery technologies for 4-1BBL) to section 5.

The description for table 2 is missing
We added a title for table 2.

Human and mouse descriptions of 4-1BB are identical, authors could fuse these in one paragraphs.
We fused that description for both in lines 77-80.

Capital letters should be used only when necessary.
Revised as requested.

Reviewer 4 Report

In this manuscript, the authors describe the implication of 4-1BBL in anti-tumor immunity and its (pre)clinical applications for immunotherapy. Since cancer immunotherapy is still confronted by major obstacles in different tumor types, strategies to enhance its effectiveness are of utmost importance. 

However some questions arise when reading this literature review.

  • First of all, the quality of English language is very poor, making the manuscript very difficult to read. The authors should seek advice for the correct use of English language.
  • In lines 38-40, the authors mention that in anti-tumor research immunity should be induced against several pathogens including tumor cells. In my opinion this is not correct. Why would would you need a response to a pathogen (except for some tumor-inducing viruses such as HPV)? Tumor cells are not pathogens. The authors should rephrase.
  • In lines 58-64, the authors mention that the immunogenicity of TAA is hampered by immune evasion by the tumor cells (such as loss of MHC) and that strategies of in vivo delivery have been developed to generate more potent responses. How can in vivo delivery of antigens prevent tumor immune evasion?
  • A general comment on the whole manuscript is that statements are often made without much explanation, so that it is difficult for the reader to comprehend. For example, in lines 179-181 about elimination of immune suppression, it is not clear in which tumor model, in which context the experiment was carried out. Also in lines 181-183, the statement about immunogenic cancer cell death as vaccines is mentioned out of the blue, without any explanation. 
  • Section 4 about the tumor microenvironment can be omitted from the manuscript in my opinion. It is not detailed enough to be comprehensive of this field and the role of 4-1BBL is mostly immunostimulatory. It is better to briefly mention the concept and then refer to a good review about this topic. 
  • In lines 214-215, the authors mention that stimulating a specific immune response can counteract immunosuppression. In my opinion this is not true, it can even exacerbate tumor immune suppression by over-stimulation. 
  • Table 2 has no title
  • In table 2, what does TMZ mean?

Author Response

We appreciate the time and effort that you have dedicated to providing your valuable feedback on our manuscript.

Comments

Point 1: First of all, the quality of English language is very poor, making the manuscript very difficult to read. The authors should seek advice for the correct use of English language.
To address this matter, we have requested the English editing service.

Point 2: In lines 38-40, the authors mention that in anti-tumor research immunity should be induced against several pathogens including tumor cells. In my opinion this is not correct. Why would would you need a response to a pathogen (except for some tumor-inducing viruses such as HPV)? Tumor cells are not pathogens. The authors should rephrase.
We reword (lines 38-40).

Point 3: In lines 58-64, the authors mention that the immunogenicity of TAA is hampered by immune evasion by the tumor cells (such as loss of MHC) and that strategies of in vivo delivery have been developed to generate more potent responses. How can in vivo delivery of antigens prevent tumor immune evasion?

In this review, is cited a systematic review showing different adjuvants associated with HPV proteins in DNA vaccines and the mechanisms by which they increase the antigen-specific immune response [1] and others articles that support this. Some of these studies are listed below:

CpG ODN with the HPV16 E7 long peptide-based vaccine has antitumor activity against E7 and also increases the expression of the MHC-I molecule. When tumors grow dramatically and are undetectable for CTL, CpG ODN adjuvant may increase the chance of tumor recognition in the early stage and therefore decrease the risk of tumor reactivity [2]. 

Combination of GM-CSF with peptide vaccines enhances antigen uptake via cross-presentation. Nanoparticles with Tat-HPV16 E7/pGM-CSF have been reported to enhance the CTL response and decrease tumor growth. This type of vaccine formulation formed the 20-80 nm nanoparticles, and greatly improved epitope-specific immunity both ex vivo and in vivo [3].

Chimeric form of 4-1BBL (SA-4-1BBL) has potent costimulatory activity in soluble form. Vaccination with SA-4-1BBL and a mutated form of HPV-16 E7 protein resulted in E7 protein-specific CD4+ and CD8+ T cell primary and long-term memory responses [4]. 

Most immunotherapeutic agents currently in development could broadly be categorized into: (1) drugs targeting the tumor immune evasion via blockade of negative regulatory signals (e.g., co-inhibitory checkpoints and tolerogenic enzymes) and (2) agents that directly stimulate immunogenic pathways (e.g., agonists of costimulatory receptors). Additional immunostimulatory strategies include enhancers of antigen presentation (e.g., vaccines), the use of exogenous recombinant cytokines, oncolytic viruses, and cell therapies using native or modified antigen-competent immune cells [5]. 

Point 4. A general comment on the whole manuscript is that statements are often made without much explanation, so that it is difficult for the reader to comprehend. For example, in lines 179-181 about elimination of immune suppression, it is not clear in which tumor model, in which context the experiment was carried out. Also in lines 181-183, the statement about immunogenic cancer cell death as vaccines is mentioned out of the blue, without any explanation. 

We add a more detailed explanation for this section (lines 179-185).

Point 5. Section 4 about the tumor microenvironment can be omitted from the manuscript in my opinion. It is not detailed enough to be comprehensive of this field and the role of 4-1BBL is mostly immunostimulatory. It is better to briefly mention the concept and then refer to a good review about this topic. 

The purpose of this section is only to mention the key characteristics of the tumor microenvironment to consider at the time of mention as target of the different strategies used in preclinical and clinical trials. Due to we consider of importance keep it in the manuscript.

Point 6. In lines 214-215, the authors mention that stimulating a specific immune response can counteract immunosuppression. In my opinion this is not true, it can even exacerbate tumor immune suppression by over-stimulation. 

We complement this mention (lines 307-310). In addition, we cite a reference for this statement:

Immunotherapy that activates the host immune system to reverse immunosuppression has emerged as a new generation of cancer treatment in both preclinical studies and clinical trials [6].

Point 7. Table 2 has no title.

We added a title to table 2.

Point 8. In table 2, what does TMZ mean?

Trimerized membrane-bound.

  1. Mousavi, T.; Sattari Saravi, S.; Valadan, R.; Haghshenas, M.R.; Rafiei, A.; Jafarpour, H.; Shamshirian, A. Different Types of Adjuvants in Prophylactic and Therapeutic Human Papillomavirus Vaccines in Laboratory Animals: A Systematic Review. Arch. Virol. 2020, 165, 263–284, doi:10.1007/s00705-019-04479-4.
  2. Khong, H.; Overwijk, W.W. Adjuvants for Peptide-Based Cancer Vaccines. J. Immunother. Cancer 2016, 4, 56, doi:10.1186/s40425-016-0160-y.
  3. Tang, J.; Yin, R.; Tian, Y.; Huang, Z.; Shi, J.; Fu, X.; Wang, L.; Wu, Y.; Hao, F.; Ni, B. A Novel Self-Assembled Nanoparticle Vaccine with HIV-1 Tat₄₉₋₅₇/HPV16 E7₄₉₋₅₇ Fusion Peptide and GM-CSF DNA Elicits Potent and Prolonged CD8+ T Cell-Dependent Anti-Tumor Immunity in Mice. Vaccine 2012, 30, 1071–1082, doi:10.1016/j.vaccine.2011.12.029.
  4. Sharma, R.K.; Srivastava, A.K.; Yolcu, E.S.; MacLeod, K.J.; Schabowsky, R.-H.; Madireddi, S.; Shirwan, H. SA-4-1BBL as the Immunomodulatory Component of a HPV-16 E7 Protein Based Vaccine Shows Robust Therapeutic Efficacy in a Mouse Cervical Cancer Model. Vaccine 2010, 28, 5794–5802, doi:10.1016/j.vaccine.2010.06.073.
  5. Velcheti, V.; Schalper, K. Basic Overview of Current Immunotherapy Approaches in Cancer. Am. Soc. Clin. Oncol. Educ. Book 2016, 298–308, doi:10.1200/EDBK_156572.
  6. Qi, F.; Wang, M.; Li, B.; Lu, Z.; Nie, G.; Li, S. Reversal of the Immunosuppressive Tumor Microenvironment by Nanoparticle-Based Activation of Immune-Associated Cells. Acta Pharmacol. Sin. 2020, 41, 895–901, doi:10.1038/s41401-020-0423-5.

Round 2

Reviewer 2 Report

Suitable for publication

Author Response

(The authors gave the same response as above.)
